# Single-centre, single-blinded, randomised, parallel group, feasibility study protocol investigating if mandibular advancement device treatment for obstructive sleep apnoea can reduce nocturnal gastro-oesophageal reflux (MAD-Reflux trial)

Saoirse O'Toole [1,2] Rebecca Moazzez,[3] Gabriella Wojewodka [4] Sebastian Zeki,[5] Jafar Jafari,[5] Katherine Hope,[6] Andrew Brand,[7] Zoe Hoare [7] Suzanne Scott,[8] Kodchawan Doungsong,[9] Victory Ezeofor,[9] Rhiannon Tudor Edwards [9] Panagis Drakatos,[5] Joerg Steier [10,11]

For numbered affiliations see end of article.

**Correspondence to**
Dr Saoirse O'Toole;
saoirse.otoole@kcl.ac.uk

## ABSTRACT

**Introduction** Just under half of patients with obstructive sleep apnoea (OSA) also have gastro-oesophageal reflux disease (GORD). These conditions appear to be inter-related and continual positive airway pressure (CPAP) therapy, the gold standard treatment for OSA to prevent airway collapse, has been shown to reduce GORD. As the impact of mandibular advancement devices, a second-line therapy for OSA, on GORD has yet to be investigated, a feasibility study is needed prior to a definitive trial.

**Methods** This will be a single-centre, single-blinded, tertiary-care based, interdisciplinary, parallel randomised controlled study. Potential OSA participants presenting to the sleep department will be pre-screened for GORD using validated questionnaires, consented and invited to receive simultaneous home sleep and oesophageal pH monitoring. Those with confirmed OSA and GORD (n=44) will be randomly allocated to receive either CPAP (n=22) or MAD therapy (n=22). Following successful titration and 3 weeks customisation period, participants will repeat the simultaneous sleep and oesophageal pH monitoring while wearing the device. The number of patients screened for recruitment, drop-out rates, patient feedback of the study protocol, costs of interventions and clinical information to inform a definitive study design will be investigated.

**Ethics and dissemination** Health Research Authority approval has been obtained from the Nottingham 2 Research Ethics Committee, ref:22/EM/0157 and the trial has been registered on ISRCTN (https://doi.org/10.1186/ISRCTN16013232). Definitive findings about the feasibility of doing 24 hour pH oesophageal monitoring while doing a home sleep study will be disseminated via clinical and research networks facilitating valuable insights into the simultaneous management of both conditions.

**Trial registration number** ISRCTN Reg No: 16013232.

### STRENGTHS AND LIMITATIONS OF THIS STUDY

⇒ This feasibility protocol is designed to gauge patient recruitment, acceptability and signal clinical efficacy for a definitive power calculation.
⇒ The clinical trial is interdisciplinary involving dental, sleep and gastroenterology clinicians alongside patients, health psychologists, health economists and statisticians.
⇒ Feasibility for health economic outcomes is also being assessed.
⇒ Findings will inform a definitive trial rather than a clinical outcome.

## INTRODUCTION

One billion people worldwide and 8 million people in the UK (24.5% of the population) are estimated to have obstructive sleep apnoea (OSA).[1] It is the most common sleep disorder, characterised by frequent pauses in breathing (apnoeas and hypopnoeas) during sleep due to airway blockage caused by intermittent relaxation of the throat muscles. Sleep becomes broken, so in the daytime patients reported of sleepiness, fatigue and have other symptoms such as difficulty concentrating, memory impairment, feeling irritable and depressed.[2] Their own and their bed partner's quality of life is reduced.[2] Patients are at a higher risk of type 2 diabetes, high blood pressure, heart disease, strokes and dying of all causes compared with people without OSA. This risk increases with OSA severity; with severe OSA, the risk of cardiac

events and strokes is doubled and of cardiac death is tripled.[3] People with untreated OSA also put the public in danger as they have more road traffic accidents than people without OSA and compared with themselves once on treatment.[4 5] The societal level-costs of undiagnosed OSA in the USA alone amounted to over $150 billion per year.[6] In Italy, OSA causes an economic burden ranging from €10.7 to €32 billion per year.[7]

Gastro-oesophageal reflux disease (GORD), defined by the montreal consensus as 'a condition that develops when the reflux of stomach contents causes troublesome symptoms and/or complications' with oesophageal/typical and extraoesophageal/atypical symptoms,[8] is also common in the UK. There is a strong association between OSA and GORD, partly due to common risk factors such as obesity, high alcohol consumption and diet and also due to the impact that apnoeic events have on the lower oesophageal sphincter (LOS).[9] A recent meta-analysis estimated that 45.2% of OSA patients had episodes of extra-oesophageal reflux independent of BMI.[9] Around 40–81% of patients suffering from GORD report that their symptoms occur during sleep.[10] There is a growing body of evidence that ontinuous positive airway pressure (CPAP) therapy, the gold-standard therapy for OSA,[11] can reduce levels of gastro-oesophageal disease[12–15] by maintaining a patent airway, thus reducing intrathoracic pressure differentials. CPAP therapy increases the baseline LOS barrier pressure during sphincter relaxation and decreases the duration of sphincter relaxation.[16] Mandibular advancement devices (MADs), a second line therapy, also maintain a patent airway, which may also have a similar impact on intrathoracic pressure differentials and duration of LOS relaxation. In addition, the greater compliance observed with MADs may mean that reflux is suppressed for a greater proportion of the night.[17] However, both theories remain untested to date. We do not have sufficient data to gauge an effect size for a definitive powered trial. We do not know if patients are willing to wear multiple devices and tests while trying to sleep and whether acceptability will influence the data. We are also unaware of the most appropriate quality of life measure to accurately assess patient impact and health economics. Therefore, a feasibility trial is needed prior to a definitive evaluation.

From an NHS perspective, CPAP and MADs are both cost-effective treatments.[18] CPAP therapy has been shown to generate an incremental cost-effectiveness ratio below £5000 per quality-adjusted life year (QALY) gained.[19] Using CPAP over a period of 14 years resulted in saving National Health Service (NHS) costs (nearly £1000 per patient) and generated health benefits for people with OSA, for example, reduction in stroke risk and road traffic accidents.[20]

In the UK, the yearly medication costs of GORD treatment are around £200 million for histamine receptor antagonists and £300 million for proton pump inhibitors (PPIs).[21] The cost implication of the impact of CPAP therapy or MAD therapy on GORD has yet to be evaluated.

The overall aim of this project is to conduct a feasibility study to assess patient recruitment and tolerance of the trial. The secondary aims are to calculate an estimate of the primary outcome effect to determine the sample size needed for the definitive trial if the progression criteria are met, to determine the optimum measure of quality-of-life impact for this patient cohort.

## METHODS AND ANALYSIS
### Study design
This will be a single-centre, single-blinded, tertiary care-based, interdisciplinary parallel 1:1 randomised controlled study. The trial has been registered on ISRCTN (https://doi.org/10.1186/ISRCTN16013232).

### Setting
Participants will be recruited from King's Health Partners sleep services at Guy's and St Thomas's Foundation Trust (GSTT, London, UK).

### Outcomes
*Primary outcome measures*
1. Percentage of approached patients who were screened for the trial measured using the percentage of patients eligible of those screened at ay 2 of the screening visit.
2. Percentage of eligible patients who were randomised measured using the percentage of screened patients who met the eligibility criteria versus those who were randomised at the randomisation timepoint.
3. Percentage of patients who completed the trial measured using number of patients who were randomised versus those who completed the trial at Day 2 of the final visit.

*Primary clinical outcome to signal efficacy and inform a definitive power calculation*
1. Change in percentage acid contact time pH <4 with the device in situ measured using nocturnal data from 24-hour pH monitoring at initial gastroenterology screening visit and follow-up gastroenterology assessment post-therapy.

*Secondary outcome measures*
1. Patient acceptability of the trial measured using qualitative interviews throughout the trial.
2. Hours that device is worn during sleep measured using the average number of hours worn per night as detected from the output from the therapeutic device throughout the trial period.
3. To assess the most sensitive quality of life questionnaire prior to the full trial by recording EuroQol 5-dimension 5-level (EQ-5D-5L),[22] ICEpop CAPability measure for the adult population (ICECAP-A)[23 24] and the Short Warwick-Edinburgh Mental Wellbeing Scale (SWEMWBS[25]) at baseline and post-intervention.
4. Rating of both therapies measured using a Visual Analog Scale at day 2 of final assessment visit (trial completion).

5. Number of potential participants identified by the care team with and without screening of referral letters measured using number of participants identified on clinics before the prescreening clinic and after the pre-screening.
6. Change in Reflux Symptom Index (RSI)[26] using changes detected in the RSI validated questionnaire at baseline and post intervention.
7. Change in Epworth Sleepiness Scale (ESS)[27] using changes detected in the ESS validated questionnaire at baseline and post intervention.
8. Change in cough using changes detected in the validated Leicester Cough Questionnaire[28] at baseline and post intervention.
9. To identify and measure indicative costs and outcomes and select suitable economic outcomes measured by documenting health resource use including intervention costs, appointment times and attending healthcare practitioners throughout the trial.
10. To determine the acceptability of the intervention and economic data collection methods measured using qualitative questions at the qualitative interview (various timepoints).

## Participants and recruitment

Potential participants will be identified by the direct care team either by running a search text function to screen their referral letters or by their consultation in clinic. Patients referred for investigation of OSA, who also have a previously confirmed diagnosis of GORD with a 24-hour pH study or typical heartburn and/or regurgitation >3 times per week and Reflux Disease Questionnaire questionnaire score of >50%, will be informed by their direct care team that they may be eligible to be included in the study and asked if they would like to speak to a research dentist or nurse about participation. The prescreening questions for dental examination will ask patients if they have at least 10 teeth in each jaw, no loose teeth, fillings or caps or obvious holes in your teeth. Interested participants, who meet the pre-screening criteria, will be provided with a patient information sheet of the entire trial and invited to have a dental screening during their gastroenterology appointment, followed by a combined home sleep study and 24-hour pH impedance oesophageal monitoring. It will be explained that participation in the trial will be dependent on meeting the strict inclusion criteria, but they will be reimbursed for the screening appointment.

## Inclusion/exclusion criteria

Patients will be included in the study if they are:
1. Aged 18 or over.
2. Confirmed OSA with Apnoea-Hypopnoea Index score between 10 and 50,
3. Confirmed GORD with greater than 6% of acid exposure time <pH 4 over 24 hours.
4. No previous CPAP or MAD therapy.

5. Sufficient healthy teeth to support an MAD, that is, 10 teeth in each jaw, no periodontal pockets >5, no frank cavitation or loose crowns/bridges.
6. Willing and able to provide informed consent to the study.

Patients will be ineligible if:
1. Pregnant or breast feeding.
2. Unable or unwilling to stop GORD medication 2 days prior to assessment or unable to undergo manometry and pH impedance testing.
3. Known liver disease or oesophageal/gastric varices.
4. Previous surgery or intervention for reflux such as fundoplication, which may preclude pH impedance testing.
5. Any previous treatment for oesophageal neoplasia.
6. Unable/unwilling to tolerate either a CPAP mask or an MAD,
7. Medical history likely to impact on 24-hour impedance testing, for example, bulimia nervosa,
8. Participation in other research within previous 30 days.

## Study screening procedures

Following written informed consent by a trained member of the research team, participants will undergo assessments at a screening visit to ensure eligibility is met. Participants will be informed that PPIs will need to be stopped 7 days prior and H2-receptor antagonists or antacids 48 hours prior to this visit. Participants will complete an RSI questionnaire, ESS questionnaire, Leicester Cough Questionnaire and several QoL questionnaires. A full dental examination will be conducted to ensure that participants are dentally fit to wear an MAD if allocated to this intervention. For the gastroenterology assessment, the patient will initially need to undergo high-resolution manometry (HRM). Following local analgesia of the nares the catheter will be introduced transnasally and the patient instructed to drink water through a straw while the HRM catheter is advanced to the stomach. The HRM catheter depth will be adjusted to ensure manometric visual of the upper oesophageal sphincter, the gastro-oesophageal junction and gastric pressures. Ten single swallows of 5 mL will be performed with each being 20 s apart. Each 5 mL water swallow will be assessed in accordance with Chicago classification (V.3) using Manoview software (V.3) (Sierra Scientific Instruments). The HRM catheter will then be removed.

Patients will then undergo reflux monitoring using Sandhill Scientific multichannel impedance pH catheters (ZANBG-44), which are inserted transnasally after applying local anaesthesia (xylocaine). The dual pH sensors of the catheter will be positioned 5 cm below and above the manometric LOS. The impedance sensors will be positioned above the LOS by 3 cm, 5 cm, 9 cm, 15 cm and 19 cm. The data will be captured by ZepHrTM recording device.

For the 24 hours while the probe is inserted, the participant will be asked to complete a food diary as food and drink will impact on their reflux. Stopping reflux

medication and the food diary are part of standard care for this procedure.

Following placement of the catheter, participants will be invited for their home sleep study. A type 3 sleep study device, the WatchPAT 200 (WP200; Itamar Medical, Caesarea, Israel) will be given in addition to comprehensive instructions on how to perform a home sleep study. A 24-hour number for technical support will be provided to the patient. After their overnight sleep study, they will return the following day to the gastroenterology department for removal of the probe and return of the WatchPAT.

The gastroenterology data will be captured by ZepHrTM recording device and data will be analysed using the BioVIEW Analysis software (V.5.7.1.0). The sleep study data will be analysed by a qualified sleep technician.

### Randomisation procedures

Participants who meet the inclusion/exclusion criteria will be allocated 1:1 using fixed-block randomisation by the purpose-built computer-generated randomisation service provided by King's Clinical Trials Unit (KCTU) to either the CPAP (n=22) or the MAD (n=22) arm. Once the participant identifier is entered into the system, an e-mail will be sent to the principal investigators and trial manager informing them of the allocation.

### Interventions

The intervention will consist of either an MAD (n=22, SomnoMed Avant, Somnomed UK) with the dental sleep medicine department or CPAP (n=22, S8/S9, ResMed, Sydney, Australia) with the sleep medicine department.

### Titration of devices and repeat assessment

The CPAP mask will be delivered as standard of care and, therefore, may be different for each patient. The same CPAP therapy allocated to the participant will be used in the follow-up investigation. Some adjustment of the devices may be needed to ensure comfort and efficacy of the device. This will be done as routine care and scheduled as needed.

Following successful titration for each device and a 3-week customisation period, the 24-hour impedance monitoring and home sleep study will be repeated. As at visit 1, participants will be asked to cease their PPIs, H2-receptor antagonists or antacids. A dental examination will be carried out by the research dentist to ensure there have been no changes in the oral cavity. Participants will be asked to review their diet diary from the day they first did the test and repeat what they ate or drank as closely as possible for the day. That afternoon the participant will attend the oesophageal physiology laboratory. Participants will repeat the RSI questionnaire, ESS questionnaire, Leicester Cough Questionnaire and a QoL questionnaire. The pH impedance testing and Watch-PAT testing will be repeated as described above. Compliance levels for the MAD and CPAP therapy on the same night will be obtained.

On returning the next day to have their pH probe removed and to return the Watch-PAT, participants will be asked to complete a questionnaire about their views in taking part in the study.

### Blinding

This is a single-blinded study. The statistician is the only member of the team who can be blinded to the allocation. The database has been created such that the intervention is not known to the statistician when performing the analysis. The patient and clinician cannot be blinded to the group allocation due to the nature of the therapeutic devices used. However, the objective, computer-generated clinical measurement outcome in for the definitive trial is less likely to be subject to bias.

### Data collection and management

Clinical, demographic and questionnaire data will be collected on case report forms. Clinical data from manometry, impedance testing and sleep study will be obtained through patient electronic records and recorded on case report forms. Paper forms will be used, and data will be recorded on electronic case report forms held in a database using MACRO V.4.0 (Informed) and maintained by the KCTU. Data checks will be performed by the trial manager and the trial statistician. The statistician and principal investigator will have access to the full trial data set at the end of the trial.

### Qualitative interviews

To enhance our understanding of acceptability of recruitment processes, screening procedures and of taking part in the trial, we will conduct telephone interviews with participants (approximately n=16) over the course of the trial. We will use a sampling matrix to include patients who elected not to take part, eligible patients who were screened but did not take part in the trial, patients who could not tolerate the device or discontinued the trial for other reasons, and those who completed the trial.

Using the Theoretical Framework of Acceptability[29] to underpin the interview guide, participants will be asked about different domains of acceptability (eg, affective attitude, burden, coherence, perceived effectiveness) for each element of the trial, including recruitment, randomisation, appointments and procedures, use of devices. Participants will also be asked about barriers to trial participation, potential improvements and how to create advocacy for the trial among stakeholders. Participants will receive an additional £25 if they take part in this 25 min interview. The interviews will be audio-recorded, anonymised and professionally transcribed verbatim, in preparation for framework analysis[30] whereby a thematic framework is developed and applied to the transcripts, allowing systematic analysis and interpretation of the qualitative data. Data will be managed in NVivo.

### Health economic measures

During the feasibility phase, the best way of collecting relevant service costs, and patient borne costs, from both

an NHS and a wider societal perspective will be investigated to inform a future full trial context.[31–33] We will consider to what extent this intervention constitutes a complex intervention, drawing on guidance from the UK Medical Research Council for the guidance for complex interventions.[34] In the context of the patient, we will pilot the (EQ-5D-5L),[22] ICECAP-A[23 24] and the SWEMWBS,[25] which could be used in a full RCT to calculate cost per QALYs and wider outcomes. We will determine if these are sufficiently sensitive to measure change in the patient group. We will also determine the appropriate deterministic sensitivity analysis in a full economic analysis in this patient group. We will pay attention to distributive cost-effectiveness analysis opportunities in a full trial.[35]

Our base-case analysis will explore the likely total costs (intervention costs and medical costs) and outcomes (health resource service use and health-related quality of life) of using MADs compared with CPAP therapy in people with OSA who have nocturnal GORD in a future full trial. We will also explore if these outcome measures accurately reflect the conditions of the patients.

We will write a report to the funder and papers for peer-review publication following the Consolidated Health Economic Evaluation Reporting Standards statement[36] and the Assessment of the Validation Status of Health-Economic Decision Model.[37] The health economic feasibility study will be reported in accordance with the Consolidated Standards of Reporting Trials extension.[38]

A full schedule of events is given in online supplemental table 1 and the patient flow diagram with feasibility outcomes is illustrated in figure 1.

### Adverse events

All adverse events (AEs) will be recorded from the time of randomisation. AEs will be classified according to severity and whether related to the study intervention. All serious AEs will be reported immediately by the chief investigator and, no later than 24 hours, to the research & development office (sponsor) and main research ethics committee.

### Sample size

There are several studies investigating change in per cent acid exposure from baseline while wearing the CPAP device. Sample sizes suggested from current data based on change in total acid were between 86 to 168 per group (excluding any attrition).[12 39] Using the smallest effect size published (Tawk et al[12]), assuming a one-sided non-inferiority design with a margin of 4.15 (per cent total acid) together with an SD of 10.8 with a power of 90% and alpha of 2.5% for a definitive study, this would require a sample of 288 (without attrition). Considering the likelihood of the main study finding an effect of this size within a 80% CI, then an approach taken by Cocks and Torgerson[40] requires a feasibility sample equivalent to approximately 9% of the proposed definitive sample requiring 26 participants. Accommodating an overall attrition rate, including tolerance to the devices, of 40%

requires a sample of 44 to be randomised. A sample of 44 will also give us a 95% CI of +/−14% around the attrition rate.

### Statistical analysis

Baseline characteristics will be summarised for all participants within the trial arms.

The feasibility outcomes include:

1. Percentage of screened patients who were eligible for the trial.
2. Percentage of eligible patients who agreed to participate.
3. Percentage of patients who completed the trial.

We will then assess one clinical outcome to assess signal of efficacy, which is the change in percentage acid contact time pH <4 with the device in situ. Participants' uptake of and adherence to both CPAP and MAD, as well as follow-up rates, will be summarised and presented as percentages. Although determining differences in clinical outcomes between the arms is not the primary purpose of this feasibility study, comparisons will be undertaken to investigate the feasibility of studying these outcomes and to calculate potential estimates and 95% CIs. As recommended in guidelines for good practice for the analysis of pilot studies,[41] the focus of the results will be on the estimates of the treatments rather than statistical significance and as such no hypothesis testing will be undertaken. Differences between the two comparison groups will be presented in the form of an unadjusted mean difference for continuous outcomes, and an OR for binary outcomes, with their associated 95% CIs. These comparisons will be made on an intention to treat basis with consideration given to per protocol analysis as sensitivity. While every effort will be made to minimise missing data, assessment of the levels of missing data will indicate suitability of measures to be continued into the definitive trial.

### DISCUSSION

Given the increasing prevalence of both OSA and GORD, it is important to further investigate the inter-relationship between the conditions and the impact of OSA treatment on GORD. While there is evidence that CPAP reduces GORD, the effect size is unclear, and the impact of MADs on GORD is entirely un-investigated.

This feasibility trial is designed as a first step to provide this important information for patients deciding on treatment choices and the prescribing sleep clinician. This trial will give us insights into the feasibility for testing patients, patient experience, the mechanism of action and the overall effect. There were several items for consideration when developing the study design. First, we chose to use a level 2 sleep study rather than polysomnography. This was because a level 2 device is sufficient for diagnosis of OSAS provided no other respiratory/sleep pathologies are suspected. At the time of recruitment, all participants had been assessed by their managing

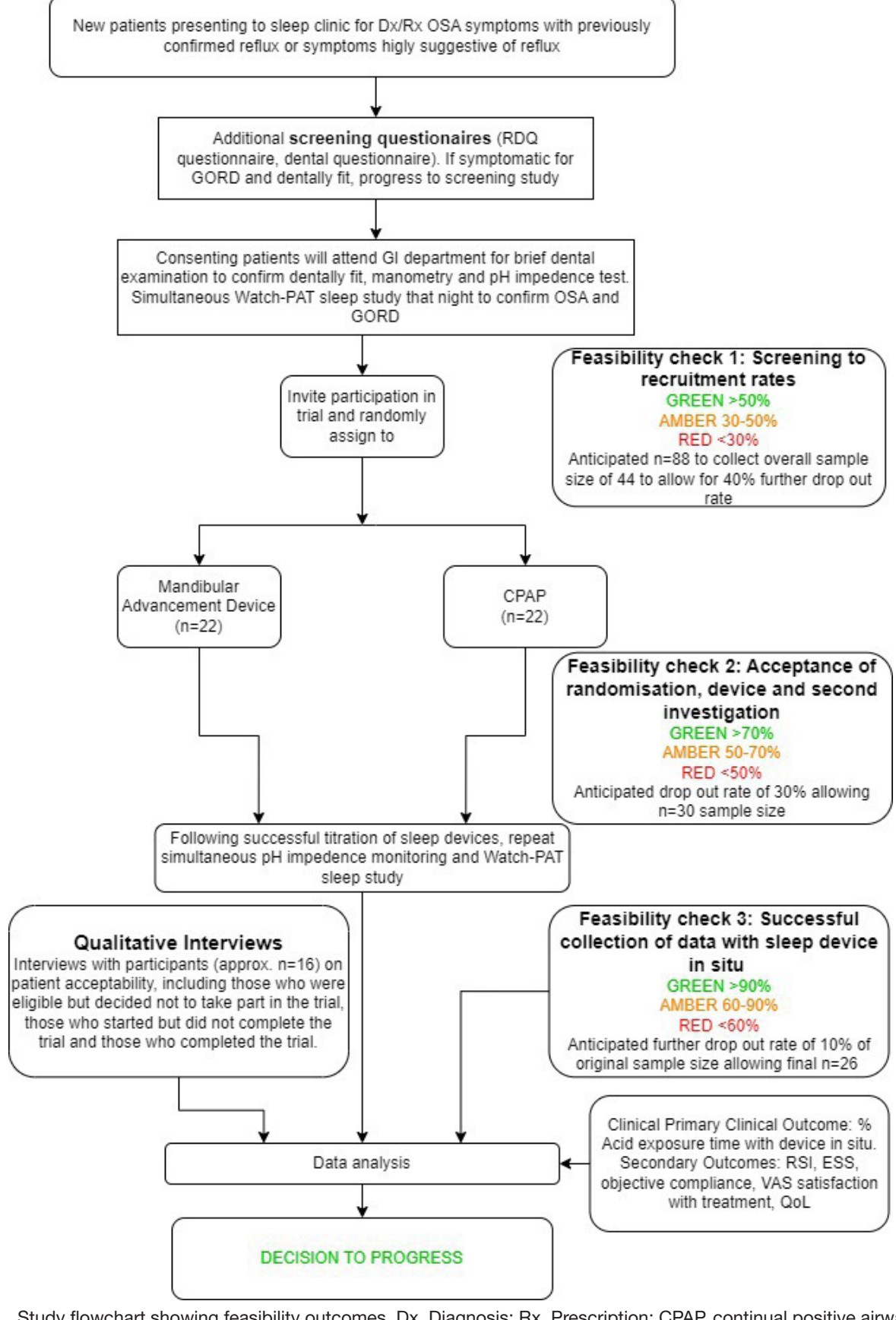

**Figure 1** Study flowchart showing feasibility outcomes. Dx, Diagnosis; Rx, Prescription; CPAP, continual positive airway pressure; ESS, Epworth Sleepiness Scale; GORD, gastro-oesophageal reflux disease; OSA, obstructive sleep apnoea; RDQ, Reflux Disease Questionnaire; RSI, Reflux Symptom Index; PAT, Peripheral Arterial Tonometry; VAS, Visual Analog Scale; QoL, Quality of Life.

consultant and diagnosed as having suspect OSAS. The primary outcome of this study is a gastroenterology outcome and AHI is used solely as the inclusion criteria. Furthermore, the final assessment night for the patient will necessitate them to wear a sleep study device, a nasal pH monitor and their MAD or CPAP therapeutic intervention. We thought this would be easier for the patient if done at home. Finally, we would like the definitive trial to be a multicentre trial. Not all centres will have capacity for polysomnography. For these reasons, we deemed that a Watch-PAT was appropriate. We have also considered items such as the interference of the nasal catheter with the CPAP mask. This has been extensively discussed with our PPI group, some of whom have already undergone 24-hour pH monitoring while wearing their CPAP. The mask seal can be maintained with strategic placement of the nasal catheter and tape. There is also precedent for this in previous studies.[12 16] However, the feasibility of this will be explored.

We will also explore health economic measures to best capture the cost of services versus the value obtained for patients and healthcare providers. From a UK NHS perspective, both CPAP and MADs are reported to be similarly cost-effective treatments.[18] This may change once their efficacy on the management of nocturnal gastro-oesophageal reflux is assessed. Investigation into the health economic feasibility for other funding systems is beyond the scope of this initial feasibility paper. However, we will be informing on the best measures to capture health economic outcomes, which can then be used in different healthcare funding environments.

## Patient and public involvement statement

The trial was designed to generate evidence for patients with both OSA and GORD who are deciding on their treatment options. Our patient coapplicant is highly active within the OSA community with both feedback and dissemination means at a national level (Hope2Sleep Charity). We have an active PPI group who were also involved in the review and development of study protocol and patient information sheets. Patient advocacy is an important part of our oversight committee and our patient coapplicant is in charge of ensuring that the patient perspective is presented throughout the duration of the trial lifecycle. All findings from the study will be made available for dissemination to the public.

## ETHICS AND DISSEMINATION

Health Research Authority approval has been obtained from the Nottingham 2 Research Ethics Committee, ref:22/EM/0157 and the trial has been registered on ISRCTN (https://doi.org/10.1186/ISRCTN16013232) with access to the full protocol. Definitive findings about the feasibility of doing 24-hour pH oesophageal monitoring while doing a home sleep study will be disseminated via clinical and research networks facilitating

valuable insights into the simultaneous management of both conditions.

**Author affiliations**
[1]Centre for Clinical, Oral and Translational Sciences, King's College London, London, UK
[2]School of Medicine, University College Dublin, Dublin, Ireland
[3]Restorative Dentistry, University of the Pacific Arthur A Dugoni School of Dentistry, San Francisco, California, USA
[4]Oral and Clinical Research Unit, King's College London, London, UK
[5]Oesophageal Physiology Laboratory, Guy's and St Thomas' Hospitals NHS Trust, London, UK
[6]Hope2Sleep, Hull, UK
[7]NWORTH (North Wales Organisation for Randomised Trials in Health), School of Medical and Health Sciences, Bangor University, Bangor, UK
[8]Queen Mary University of London Wolfson Institute of Population Health, London, UK
[9]Centre for Health Economics and Medicines Evaluation, Bangor University, Bangor, UK
[10]Lane Fox Unit/Sleep Disorders Centre, Guy's & St Thomas' NHS Foundation Trust, London, UK
[11]School of Medicine, King's College London, London, UK

**Acknowledgements** Thanks to Dr Catherine Lawrence for reading support for RTE.

**Contributors** SO'T, RM, SZ, JJ, GW, KH, AB, ZH, SS, RTE, VE, KD, PD and JS contributed to the conception and design of the study. SOT drafted the manuscript. All authors critically revised the manuscript. All authors gave their final approval and agree to be accountable for all aspects of the work.

**Funding** This project is funded by the National Institute for Health and Care Research (NIHR) under its Research for Patient Benefit (RfPB) Programme (Grant Reference Number NIHR202744). The views expressed are those of the authors and not necessarily those of the NIHR or the Department of Health and Social Care. This study is sponsored by Guy's and St Thomas's Trust and King's College London. The funder and sponsor did not have any authority over the intellectual development and decision to submit this paper for publication.

**Competing interests** None declared.

**Patient and public involvement** Patients and/or the public were involved in the design, or conduct, or reporting, or dissemination plans of this research. Refer to the Methods section for further details.

**Patient consent for publication** Not applicable.

**Provenance and peer review** Not commissioned; externally peer reviewed.

**ORCID iDs**
Saoirse O'Toole http://orcid.org/0000-0002-2144-1847
Gabriella Wojewodka http://orcid.org/0000-0003-4635-4800
Zoe Hoare http://orcid.org/0000-0003-1803-5482
Rhiannon Tudor Edwards http://orcid.org/0000-0003-4748-5730
Joerg Steier http://orcid.org/0000-0002-1587-3109

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
