## [Reviewer comments · BMJ Open]

ARTICLE DETAILS

TITLE (PROVISIONAL)	A Single-Centre, Single-blinded, Randomised, Parallel group, Feasibility Study Protocol Investigating if Mandibular Advancement Device Treatment for Obstructive Sleep Apnoea can reduce Nocturnal Gastro-Oesophageal Reflux (MAD-Reflux trial).
AUTHORS	O'Toole, Saoirse; Moazzez, Rebecca; Wojewodka, Gabriella; Zeki, Sebastian; Jafari, Jafar; Hope, Katherine; Brand, Andrew; Hoare, Zoe; Scott, Suzanne; Doungsong, Kodchawan; Ezeofor, Victory; Edwards, Rhiannon; Drakatos, Panagis; Steier, Joerg

VERSION 1 – REVIEW

REVIEWER	Segu, Marzia University of Parma
REVIEW RETURNED	18-Jun-2023

GENERAL COMMENTS	Very well designed research. Did you use the polysomnography to diagnose OSAS?
---

REVIEWER	Chapman, Julia University of Sydney, CIRUS, Centre for Sleep and Circadian Research, Woolcock Institute of Medical Research
REVIEW RETURNED	04-Jul-2023

GENERAL COMMENTS	General Comments: The authors present here an interesting study of a previously under-researched area looking at the impact of treating obstructive sleep apnea to improve symptoms of gastro-oesophageal reflux disease. The authors pose a feasibility trial design to compare the standard treatment for OSA (continuous positive airway pressure) with mandibular advancement devices. I appreciate that this trial has been ethically approved and registered, and I understand that my comments on the trial design itself may not be able to be altered at this stage. I would still be interested to hear from the authors their rationale for making certain study design choices. Can the authors confirm whether this protocol has been written in accordance with the SPIRIT statement? There is some switching between US spelling and UK spelling throughout – can this be made consistent please? Specific Comments: Title: I would suggest a more descriptive title as suggested by SPIRIT, including the comparator and population. Abstract: When the authors say “a feasibility study is needed” at the end of the first paragraph, I think there would be benefit from first saying that no trial has examined this, so a clinical trial is required. And first, this feasibility study is proposed to confirm study design
---

	parameters and also enable pilot data that would enable powering of a larger study. It is not that the feasibility study itself is needed, the trial is needed and this feasibility study is the first step to that. Introduction: Can the authors provide a reference for the claim that MADs show greater compliance than CPAP? In the section regarding cost effectiveness, it would also be of interest to know more information about the health system within which this study is being conducted, as international readers will have quite different sleep apnea therapy funding environments than the NHS. Some consideration to the feasibility of implementing this in different healthcare funding environments would be of interest. Methods: Can the authors please expand the names of the EQ5D, ICE CAP and SWEMWBS questionnaires and provide references when first mentioned (Page 5, Outcomes)? Can references be provided for the RSI, ESS, Leicester Cough questionnaires? The authors have alluded to this in the introduction, but how will the authors evaluate whether wearing a nasal tube for pH monitoring interfere with the effectiveness of either CPAP or MAD? It is possible that the nasal tubing may affect CPAP more than MAD? How might this impact on the primary clinical efficacy variable. Can the authors further detail on the method of allocation concealment? How will the participants be randomised? Will this be done electronically? Will there be any way for the coordinators to know prior to allocation which allocation is coming next? Is there more detail that can be provided about the CPAP use in lab vs at home? Will it be the same device as home use or will it be different for in lab assessment? E.g. will full-face masks need to be worn? As part of the feasibility and health economic assessment it would be of interest to evaluate the number of participants who were pre-screened out due to contraindications for MAD therapy (e.g. inadequate dentition) as this would form part of the overall effectiveness equation. Can the authors expand the term MRC please (page 10, Health Economic Measures)? And R&D, REC (page 10, adverse events)? Discussion: The discussion is very limited and does not discuss many implications of the chosen trial design in a broader context. Is some expansion here possible? There is some evidence that dental erosion is a common comorbidity of GORD. Can the authors provide comment on the available evidence and the potential impact this may have on the feasibility of MAD use in patients with GORD? Can the authors comment on the use of a Level 3 sleep study device and the impact that may have on OSA assessment? Would Level 1 or 2 sleep monitoring have been feasible in this study?
--	--

VERSION 1 – AUTHOR RESPONSE

Reviewer: 1
Dr. Marzia Segu, University of Parma

Comments to the Author:

Very well designed research.

Did you use the polysomnography to diagnose OSAS?

Thank you for reviewing it. We chose to use a WatchPAT and not polysomnography to diagnose OSAS for the following reasons:

1. It has been shown that a type 2 device is sufficient for diagnosis of OSAS provided no other respiratory/sleep pathologies are suspected. At the time of recruitment, all participants had been assessed by their managing consultant and diagnosed as having suspect OSAS.
2. The primary outcome of this study will ultimately be a gastroenterology outcome. AHI is used solely as the inclusion criteria.
3. The final assessment night for the patient will necessitate them to wear a sleep study device, a nasal pH monitor and their MAD or CPAP therapeutic intervention. We thought this would be easier for the patient if done at home.
3. Ultimately, we would like the definitive trial to be a multi-centre trial. Not all centres will have capacity for polysomnography.

This has been added to discussion, page 12

Reviewer: 2

Dr. Julia Chapman, University of Sydney

Comments to the Author:

General Comments:

The authors present here an interesting study of a previously under-researched area looking at the impact of treating obstructive sleep apnea to improve symptoms of gastro-oesophageal reflux disease.

The authors pose a feasibility trial design to compare the standard treatment for OSA (continuous positive airway pressure) with mandibular advancement devices.

I appreciate that this trial has been ethically approved and registered, and I understand that my comments on the trial design itself may not be able to be altered at this stage. I would still be interested to hear from the authors their rationale for making certain study design choices.

Can the authors confirm whether this protocol has been written in accordance with the SPIRIT statement?

A Spirit checklist in addition to the CONSORT checklist has been attached.

There is some switching between US spelling and UK spelling throughout – can this be made consistent please?

The document has been revised for consistent spelling.

Specific Comments:

Title: I would suggest a more descriptive title as suggested by SPIRIT, including the comparator and population.

The title has been altered in adherence with the SPIRIT guidelines

Abstract: When the authors say “a feasibility study is needed” at the end of the first paragraph, I think there would be benefit from first saying that no trial has examined this, so a clinical trial is required. And first, this feasibility study is proposed to confirm study design parameters and also enable pilot data that would enable powering of a larger study. It is not that the feasibility study itself is needed, the trial is needed and this feasibility study is the first step to that.

The abstract introduction has been altered to reflect your clarification. Page 1

Introduction:

Can the authors provide a reference for the claim that MADs show greater compliance than CPAP?

A systematic review and meta-analysis reference has been added. Middle of first paragraph, page 3

In the section regarding cost effectiveness, it would also be of interest to know more information about the health system within which this study is being conducted, as international readers will have quite different sleep apnea therapy funding environments than the NHS. Some consideration to the feasibility of implementing this in different healthcare funding environments would be of interest.

Investigation of the health economic feasibility for other funding systems is beyond the scope of this initial feasibility paper. However, we will be informing on the best measures to capture health economic outcomes which can then be used in different healthcare funding environments. This has been added to discussion, final paragraph, page 11.

Methods:

Can the authors please expand the names of the EQ5D, ICE CAP and SWEMWBS questionnaires and provide references when first mentioned (Page 5, Outcomes)?

These have been added, please see page 5.

Can references be provided for the RSI, ESS, Leicester Cough questionnaires?

These have been added, please see page 5.

The authors have alluded to this in the introduction, but how will the authors evaluate whether wearing a nasal tube for pH monitoring interfere with the effectiveness of either CPAP or MAD? It is possible that the nasal tubing may affect CPAP more than MAD? How might this impact on the primary clinical efficacy variable.

This has been extensively discussed with our PPI group some of whom have already undergone 24hour pH monitoring while wearing their CPAP. The mask seal can be maintained, and any mask leak can be overcome with strategic placement of the nasal catheter and tape. There is also precedent for this in previous studies. This section, along with the references for precedent-setting studies, has been added to the discussion, page 11.

Can the authors further detail on the method of allocation concealment?

This has been discussed in more detail in the Blinding section on page 8.

How will the participants be randomised? Will this be done electronically? Will there be any way for the coordinators to know prior to allocation which allocation is coming next?

All randomisation is provided by the purpose-built, web-based randomisation service provided externally by the King's Clinical Trials Unit. There is no way that the coordinators can know the outcome before the Trials Unit inform us of the allocation. This has been added to page 7.

Is there more detail that can be provided about the CPAP use in lab vs at home? Will it be the same device as home use or will it be different for in lab assessment? E.g. will full-face masks need to be worn?

The CPAP mask will be delivered as standard of care and therefore may be different for each patient. The same CPAP therapy allocated to the participant will be used in the follow up investigation. This has been added to page 8.

As part of the feasibility and health economic assessment it would be of interest to evaluate the number of participants who were pre-screened out due to contraindications for MAD therapy (e.g. inadequate dentition) as this would form part of the overall effectiveness equation.

All reasons for failing the pre-screening appointment will be documented as standard procedures during a clinical trial.

Can the authors expand the term MRC please (page 10, Health Economic Measures)? And R&D, REC (page 10, adverse events)?

Acronyms have been fully expanded upon.

Discussion:

The discussion is very limited and does not discuss many implications of the chosen trial design in a broader context. Is some expansion here possible?

This has been expanded upon utilising your comments. Thank you. See page 12

There is some evidence that dental erosion is a common comorbidity of GORD. Can the authors provide comment on the available evidence and the potential impact this may have on the feasibility of MAD use in patients with GORD?

Dr. O'Toole is an international expert in dental erosion. Reduced height or altered shape of teeth resulting from dental erosion very rarely impacts on the ability of the patients to wear a Mandibular Advancement Device. Reduced number of teeth, periodontal disease (loose teeth) and active decay (holes in the teeth) are the dental conditions that may impact MAD use and are thus the ones listed in the exclusion criteria. However, all reasons for exclusion will be documented for the trial.

Can the authors comment on the use of a Level 3 sleep study device and the impact that may have on OSA assessment? Would Level 1 or 2 sleep monitoring have been feasible in this study?

This has been added to the discussion. Page 12

VERSION 2 – REVIEW

REVIEWER	Chapman, Julia University of Sydney, CIRUS, Centre for Sleep and Circadian Research, Woolcock Institute of Medical Research
REVIEW RETURNED	16-Aug-2023
GENERAL COMMENTS	Thank you for the opportunity to review after revisions. I am happy with the edits that have been made. I look forward to reading the results in the future!